# Identification of plasmon-driven nanoparticle-coalescence-dominated growth of gold nanoplates through nanopore sensing

Bintong Huang[1,3], Longfei Miao[1,3], Jing Li[1], Zhipeng Xie[1], Yong Wang [2], Jia Chai[1] & Yueming Zhai [1✉]

The fascinating phenomenon that plasmon excitation can convert isotropic silver nanospheres to anisotropic nanoprisms has already been developed into a general synthetic technique since the discovery in 2001. However, the mechanism governing the morphology conversion is described with different reaction processes. So far, the mechanism based on redox reactions dominated anisotropic growth by plasmon-produced hot carriers is widely accepted and developed. Here, we successfully achieved plasmon-driven high yield conversion of gold nanospheres into nanoplates with iodine as the inducer. To investigate the mechanism, nanopore sensing technology is established to statistically study the intermediate species at the single-nanoparticle level. Surprisingly, the morphology conversion is proved as a hot hole-controlled coalescence-dominated growth process. This work conclusively elucidates that a controllable plasmon-driven nanoparticle-coalescence mechanism could enable the production of well-defined anisotropic metal nanostructures and suggests that the nanopore sensing could be of general use for studying the growth process of nanomaterials.

[1] The Institute for Advanced Studies (IAS), Wuhan University, Wuhan 430072, People's Republic of China. [2] Shanghai Institute for Advanced Study, Institute of Quantitative Biology, College of Life Sciences, Zhejiang University, Hangzhou 310027, People's Republic of China. [3]These authors contributed equally: Bintong Huang, Longfei Miao. ✉email: yueming@whu.edu.cn

Since the first report of plasmon-induced conversion from silver (Ag) nanospheres to Ag nanoprisms in 2001[1], plasmon-driven synthesis has garnered immense interest because it can break the symmetry of spherical plasmonic metal nanoparticles for the fabrication of anisotropic nanostructures[2–8]. However, after almost two decades, the mechanism governing this evolution is still subject to debate. Two growth pathways are mainly invoked to explain such shape conversion. One mechanism describes that multiply twinned Ag nanospheres could be oxidized back to Ag ions by oxygen, supplying Ag precursors for the growth of more stable planar twinned nanoprisms, similar to Ostwald ripening process[9–13]. The other explanation assumed that Ag nanoprisms may form via oriented attachment of smaller Ag nanoprisms or Ag nanospheres, a particle-coalescence process[14–17]. Accumulated evidence has supported the possibility and importance of the first mechanism, because plasmon-driven anisotropic growth can be achieved on quasi-spherical plasmonic nanoseeds using Ag or even gold (Au) ions as precursors. Meanwhile, multiply twinned Ag nanostructures are demonstrated very sensitive to be oxidized into ions by oxygen[11,18]. In contrast, for the hypothesis of plasmon-driven nanoparticle-coalescence growth pathway, the solid evidence is extremely rare. Generally, metallic nanoparticle aggregation and coalescence are uncontrollable and induce unwanted polydispersity[19–21]. It is usually utilized for constructing highly porous gel[22–24] or optical sensing systems[25,26], but principally avoided in nanostructure manufacturing. Furthermore, it is very challenging to prove the coalescence-driven growth pathway from the point of methodology for real light-driven condition.

Recently, the advanced technique to study the colloidal nanocrystal growth is the liquid phase transmission electron microscopy (LP-TEM), which allows direct visualization of the growth process[27–29]. However, this technique still has its own limitations for mechanism studies of coalescence-driven crystal growth in some real reaction conditions, such as under light illumination. The key issues mostly arise from the high-energy electron-beam irradiation during LP-TEM characterization, which can generate free radicals, heat, and decompose of surfactant, inducing the self-adjustment of the nanoparticles through collisions or self-assembly to decrease the surface energy[12,30–33]. For example, repeated oriented attachment of tiny metal nanoparticles could be triggered by electron beam irradiation leading to multiply twinned structures during in-situ LP-TEM observation[34]. For the ex-situ TEM observation, an indeterminate number of nanoparticles may aggregate during the sample drying. Additionally, localized information is always obtained, probably causing artifacts, especially for the complex samples. These effects make it very challenging to elucidate the exact growth pathway in real reaction conditions.

Here, we successfully achieved the high yield plasmon-mediated Au nanospheres-to-nanoplates conversion using iodine (I$_2$) as the inducer. To elucidate the growth pathway, we used nanopore sensing technology in combination with TEM tracking characterization to capture the intermediate species during the nanostructure evolution. This method allows us to obtain valuable statistical information on the shape of nanoparticles at the single-nanoparticle level. Our results suggested that the high yield morphology conversion is a hot hole-controlled nanoparticle-coalescence-dominated growth process. This work conclusively proved that nanoparticle-coalescence could generate well-defined anisotropic metal nanostructures. Furthermore, our work suggested that the nanopore sensing technique could be generally used for studying the growth process of nanostructures.

## Results

### Plasmon-driven conversion of Au nanospheres to nanoplates.
In a typical experiment (Fig. 1a), 5 ml of traditional citrate-protected Au nanospheres solution was mixed with 170 μl of aqueous iodine (I$_2$, 0.5 mM), and then irradiated with $500 \pm 10$ nm light (52 mW cm$^{-2}$) for 540 min. A series of color changes, from the initial orange-red to blue-green, were observed (Fig. 1b). The TEM characterizations showed that the initial pseudo-spherical Au nanoparticles with an average size of about $5.72 \pm 0.65$ nm (Fig. 1c and Supplementary Fig. 1) were gradually converted into the plate-like structures in high yield (exceeded 90%, Fig. 1d, e) under light irradiation. The conversion process was also characterized by UV-Vis spectra (Fig. 1f). In the early stage, the intensity of the surface plasmon resonance (SPR) of Au nanospheres ($\lambda_{max} = 512$ nm, peak I) immediately decreased after I$_2$ was added into the solution. Then, the intensity of peak I reached its maximum after about 120 min of illumination, accompanied by the appearance of a new peak II. Finally, the intensity of peak I gradually decreased and almost disappeared after 540 min of irradiation. The intensity of peak II kept increasing and slowly red-shifted to 750 nm, which was attributed to the SPR feature of Au nanoplates. The photochemical conversion process also showed excellent morphological specificity and controllability in systems with 2× and 10× magnification (Supplementary Fig. 3).

Because the citrate-protected Au nanospheres are quite stable under light irradiation, the morphology conversion of Au nanospheres should be triggered by the addition of I$_2$ (mainly in the form of I$_3^-$ species, discussed in Supplementary Fig. 4). The control experiments with the same amount of I$^-$ showed that no SPR peak of Au nanoplates appeared (Supplementary Fig. 5). After the addition of I$_2$, a very rapid chemical etching of Au$^0$ happened, leading to the decrease of the extinction of Au nanoparticle solution (Fig. 1f). TEM tracking of Au nanoparticles showed that the etching is selective. The etching rate of multiply twinned nanostructures is obviously faster than that of their counterparts with less twin boundaries, which is due to their different contents of unstable twin defects with strains (Supplementary Fig. 6)[4]. The similar conversion does not take place in the dark condition (even over 48 h, Supplementary Fig. 7), indicating that the light illumination is necessary to guide the conversion of Au nanospheres into nanoplates. Without light irradiation, only irregular Au nanostructures were generated with a very broad absorption in the visible light range without distinct features (Fig. 1g). We further studied the morphological changes of Au nanospheres under different wavelengths of light excitation (Supplementary Fig. 8). The absorption spectra and SEM characterizations exhibited that the appropriate wavelength close to the surface plasmon band of Au nanospheres is conducive to the conversion towards high-yield Au nanoplates, while other wavelengths promote the formation of irregular nanostructures.

As mentioned above, since I$_2$ can selectively etch Au nanospheres to provide precursors which may be reduced back on more stable Au nanostructures due to photochemical process (Supplementary Fig. 6), the plasmon-driven conversion of Au nanospheres seems following the widely accepted redox mechanism of Ag counterpart. However, during TEM characterizations, aggregations of different types of nanoparticles were always observed at different reaction times (Fig. 1h, i and Supplementary Fig. 9). More important, no obvious etching was ever observed using TEM tracking of Au nanospheres (even penta twinned structure) obtained after 2 h reaction (Supplementary Fig. 10). Additionally, the amount of citrate did not decrease detectably by $^1$H NMR spectroscopy during the conversion process (Supplementary Fig. 11), while citrate is considered as the reduction reagent during plasmon-induced synthesis[9,11]. These observations made us revisit the long-term debated question on whether the morphology conversion was caused by the oxidation and re-deposition or by the nanoparticle-coalescence mechanism. The

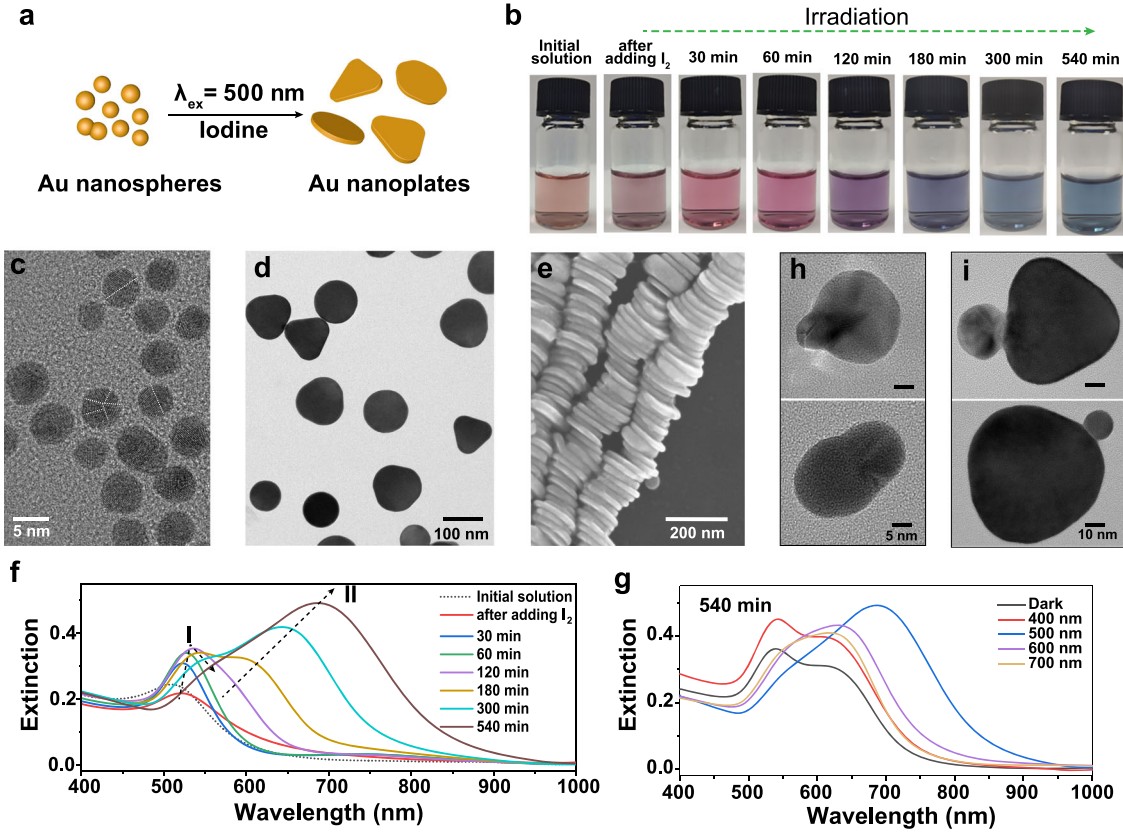

**Fig. 1 Photoinduced conversion from Au nanospheres to Au nanoplates. a** Schematic illustration of the photoinduced conversion of Au nanospheres to nanoplates in the presence of iodine with irradiation at $500 \pm 10$ nm light. **b** Photograph of the initial Au nanospheres solution, and the reaction solution under continuous light ($500 \pm 10$ nm) irradiation after adding 170 µl of $I_2$ aqueous solution (0.5 mM). **c** TEM image of the Au nanospheres. The white dotted line as a guide highlights the boundaries of the twins (HRTEM images of more Au nanospheres are shown in Supplementary Fig. 1). **d, e** TEM and SEM images of final products obtained via irradiation of the reaction solution with light for 540 min (HRTEM image of Au nanoplate is displayed in Supplementary Fig. 2). **f** Time-dependent UV–Vis spectra of the corresponding solution in (**b**). **g** UV–Vis spectra of solution after 540 min reaction in dark condition, under $400 \pm 10$, $500 \pm 10$, $600 \pm 10$, and $700 \pm 10$ nm light excitation. **h, i** TEM images of the intermediate products after 60 and 300 min conversion reaction, respectively.

advanced technique for investigating the evolution mechanism of nanostructures is direct observation using in situ LP-TEM. However, for light-mediated reactions, the challenge is to distinguish the effects of the high energy electron beam irradiation and the light excitation with specific wavelengths. Meanwhile, the limited volume in the liquid cell and the electrostatic attraction between the $Si_3N_4$ membrane and the solution could lead to an increase in the compressive stress on Au nanoparticles, inducing aggregation[33]. To address this question, we tried to capture the transformation process using ex-situ TEM. As illustrated in Supplementary Video 1, it is found that the sample preparation process may already cause unwanted aggregation and the inevitable exposure of the electron beam indeed could induce the fusion of susceptible $I_2$-treated Au nanoparticles when they are close.

**Nanopore-based analysis of Au nanostructures**. To explain the growth mechanism first requires an exact study of the Au nanostructures in the growth solution at different reaction time. Considering the limitation of the in situ LP-TEM, we investigated the morphology conversion using a nanopore sensing characterization method (Fig. 2a), which can capture the intermediate species at the single-nanoparticle level during nanostructure conversion. For the nanopore characterization, certain amount of growth solution is taken out and $KNO_3$ electrolyte was added to

support the electrochemical analysis. It is demonstrated that the present of 5 mM $KNO_3$ will not cause aggregation of Au nanoparticles (Supplementary Fig. 12). A laser-based pulled conical quartz nanopore with a suitable diameter was inserted into the reaction solution (see the "Methods" for nanopore fabrication and characterization, Supplementary Fig. 13). The ionic current was measured upon applying a voltage between the Ag/AgCl electrodes. We took an optimized salt concentration gradient strategy (50 mM/5 mM trans/cis $KNO_3$) to increase the nanopore capture rate of nanoparticles (Supplementary Fig. 14)[35–37], which also largely avoided the aggregation of the additional nanoparticles caused by the detection method (will be discussed later). We first investigated the initial solution of Au nanospheres. Supplementary Fig. 15a displays a segment of the current versus time trace with stochastic current spikes, in which selected concatenate events were extracted (Fig. 2b). The translocation events of Au nanospheres were classified into four types based on the number of peaks in each event, namely single, double, triple, and multiple peaks (including all of the events with more than three peaks). Current blockade ($I_b$) and dwell time ($t_d$) values of different peaks were extracted by peak analysis code (See the scatter plots in Supplementary Fig. 16). The mean value of $t_d$ from double-peak events is 646.1 µs in the $t_d$ histogram, shorter than twice of that from the single peak (372.8 µs). The mean value of the interval time between the double and triple-peak events is close to 200.0 µs, much smaller than $t_d$ value for single peak

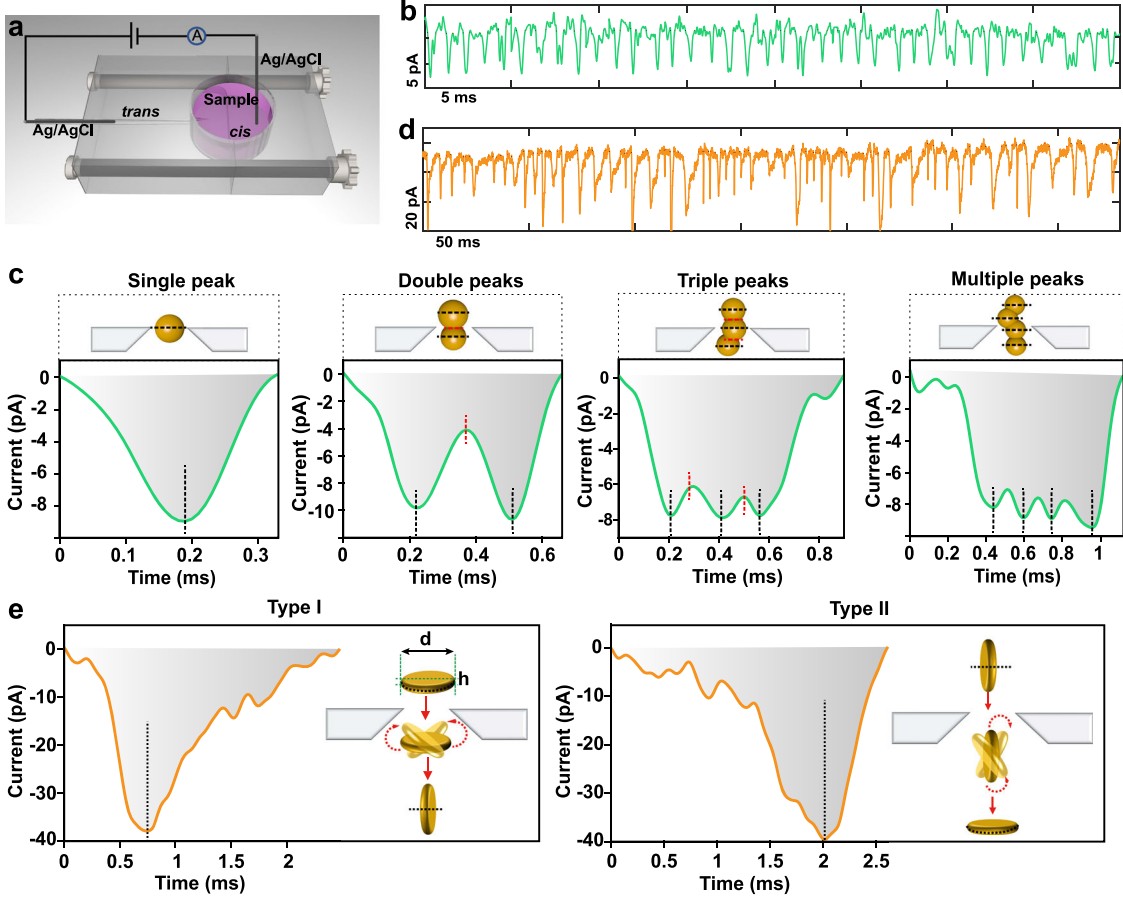

**Fig. 2 Nanopore analysis of Au nanospheres and Au nanoplates. a** Schematic of the nanopore experimental setup with a nanopipette and fluid compartments confined by polymethyl methacrylate (PMMA). The sample is added to the *cis* chamber in a 5 mM $KNO_3$ solution. 50 mM $KNO_3$ is filled in the *trans* chamber. **b** Concatenate events from Au nanospheres translocation test using a ~20 nm quartz nanopore at a 1 V bias voltage. A total of 2612 events were obtained from the Au nanospheres. **c** Typical nanopore current–time peak shape dependence of Au nanospheres coalescence states. Schematic of Au nanospheres translocation. **d** Concatenate events from Au nanoplates translocation test using a ~120 nm quartz nanopore at a bias voltage of 300 mV. A total of 761 events were obtained from the Au nanoplates. **e** Typical current–time peak shape for Au nanoplates. Schematic illustration of possible rotational tumbling of Au nanoplates in the nanopore sensing region.

(Supplementary Fig. 17). In contrast, no significant difference was found for $I_b$ in each peak from different events. We further analyzed the distribution of the $I_b/I_0$ ($I_0$ indicates the amplitude of the open pore current) of the single, double ($I_{bD1}$ and $I_{bD2}$) and triple ($I_{bT1}$, $I_{bT2}$, and $I_{bT3}$) peaks (Supplementary Fig. 18). The statistical mean value of $I_b/I_0$ for each of these peaks was very close. Based on the above analysis, we concluded that the number of peaks in each event can be used as a proxy for the coalescence state of Au nanoparticles[38]. As shown in Fig. 2c, the single peak corresponds to a single spherical-shape Au nanoparticle trans-location, accounting for 59.95% (Supplementary Fig. 16). The proportion of double peaks was 17.38%, indicating that Au nanoparticles were aggregated in a peanut-like structure, also confirmed by TEM (Supplementary Figs. 16 and 19). The $I_b$ value and the interval time between two peaks can be used to determine the size and coalescence status of individual Au nanoparticle in the peanut-like structure (see typical events and possible struc-tures in Supplementary Fig. 19). The Au nanoparticles related aggregated Au nanostructures for triple and multiple peaks are displayed in Supplementary Fig. 20.

It is necessary to validate that our detection method itself does not cause nanoparticle aggregation. Given the complex structures of the initial Au nanospheres, the sample was treated by centrifugation (7040 × *g*, 15 min) to get rid of most of large aggregates for preparing small Au nanoparticles with much better

monodispersity (Supplementary Fig. 21). Compared with the peaks of initial Au nanospheres, the single-peak events of monodispersed Au nanospheres significantly increased at 50 mM/5 mM trans/cis $KNO_3$ solution (91.97% single-peak events). Additionally, 13 nm Au nanospheres with very good monodispersity were also used for the comparation, which reached 88.68% single-peak events (Supplementary Fig. 22). These control experiments could exclude the possibility of artifact for the aggregation caused by the detection method.

The translocation events for Au nanoplates were recorded (Supplementary Fig. 23a). A typical trajectory including extracted concatenate events was shown in Fig. 2d. Unlike the Au nanospheres, the Au nanoplates exhibit anisotropic structures with a much larger crosswise orientation d than the lengthwise orientation h (Fig. 2e). In this case, the rotational tumbling of nanoplates in the nanopore results in characteristic pulse signal depending not only on its volume, but also on its orientation with respect to the electric field[39–43]. The duration to reach maximum $I_b$ is shorter in their crosswise orientation versus lengthwise orientation. Au nanoplates rotationally tumble over 90° in the nanopore sensing zone, i.e., from crosswise to lengthwise orientation (type I) or vice versa (type II), leading to two distinct asymmetric peak shapes with only one apparent maximum current blockade (Fig. 2e and Supplementary Fig. 24). The random rotation causes small fluctuation in the pulse signal

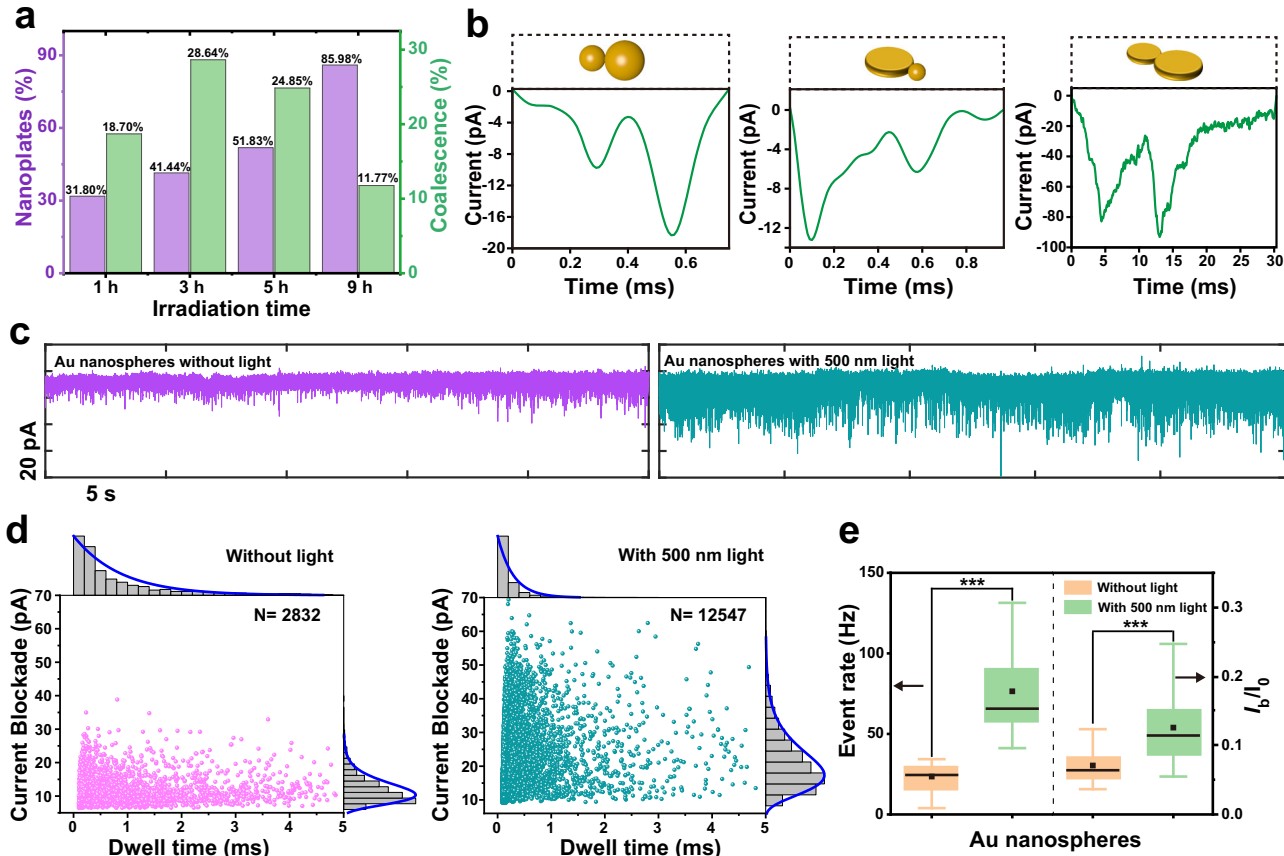

**Fig. 3 Nanopore analysis of the morphological conversion mechanism of Au nanospheres to Au nanoplates. a** The ratio of nanoplates and coalescence in the process of morphology transformation obtained from CNN prediction model and peak analysis code, respectively. 1135, 1035, 1283, and 1198 events were obtained in the 1, 3, 5, and 9 h samples, respectively. A representative current trace and event scatter plots of dwell time vs. current blockade of all the experiments are displayed in Supplementary Figs. 27 and 28. **b** Typical nanopore current–time peak shape for nanoparticle-nanoparticle, nanoparticle-nanoplate, and nanoplate-nanoplate coalescence events (left to right). **c** Current versus time trace of Au nanospheres without (left) and with 500 nm light (right). **d** Scatter plots for $t_d$ vs. $I_b$ of Au nanospheres translocation without (left) and with 500 nm light (right) from (**c**) and along the sides are the corresponding histograms. Measurements were collected using a same ~20 nm nanopore at a bias voltage of 1 V. The baseline current is about 310 pA and RMS is 2.18 pA. **e** Box plots of event rate and $I_b/I_0$ from Au nanospheres with 500 nm light and without light. Box plots show median (black line), mean (black square box), quartiles (boxes), and range (whiskers). Mann–Whitney $U$ tests, ***$p < 0.001$.

magnitude[43]. Apart from these apparent events of Au nanoplates, a small number of signals from byproduct nanospheres have been found (Supplementary Fig. 25). The $I_b$ of these signals showed overlap with the signals from nanoplates. In terms of the distinct pulse shapes between the Au nanospheres and the nanoplates, the shape of events could be used to identify the samples. Convolutional neural networks (CNNs) were then developed for the supervised classification task for identifying populations of Au nanoplates and Au nanospheres events (Supplementary Fig. 26). By parameter adjustment, we found that the model could achieve an accuracy of 95.66% on the test dataset (Supplementary Fig. 26c). We then performed a translocation study of the sample solution at different time during the reaction. The prediction of these sample events by the model revealed a gradual increase in the event content of nanoplates as the reaction progressed (Fig. 3a).

The single peak events in the 1 h sample reached 73.46%, much higher than that of initial Au nanospheres, suggesting that the aggregated nanoparticles in the initial solution can transform into larger single nanostructures (Supplementary Fig. 29a). Interestingly, we found two size distributions for both $I_{bD1}$ and $I_{bD2}$ of double peaks, indicating that the coalescence structure contains two nanoparticles with a distinct difference in diameter (Supplementary Fig. 29b). For

the 3 and 5 h sample translocation signals, coalescence events accounted for 28.64 and 24.85%, respectively (Fig. 3a). The peak shape analysis further reveals that the nanostructures aggregated and coalesced in solution, which is hard to be undoubtedly confirmed by the TEM characterization. The typical shape of the coalescence peak and the corresponding intermediate species in the conversion process are shown in Fig. 3b and Supplementary Fig. 30. In addition to the aggregations corresponding to nanoparticle–nanoparticle, nanoparticle–nanoplate and nanoplate–nanoplate coalescence events are also present in this process.

**Evolution of nanoparticle conversion process and mechanisms**. Based on the above analysis of intermediate Au nanostructures during the reaction, it is proposed that light-induced nanoparticles coalescence is the predominant route for the conversion of Au nanospheres into Au nanoplates. The light excitation plays an essential role in this morphology evolution. To explore its underlying function, the signals of Au nanoparticles through nanopore were compared between light excitation and dark condition. It is found that the capture rate of Au nanoparticles by nanopore obviously increased (Mann–Whitney $U$ test, $Z = 4.811$, $p < 0.001$) under 500 nm light irradiation. Meanwhile, the

intensities of $I_b/I_0$ also significantly enhanced (Mann–Whitney $U$ test, $Z = 62.471$, $p < 0.001$) (Fig. 3c–e). For comparison, $SiO_2$ nanoparticles, which did not absorb 500 nm light, were studied in the same condition (Supplementary Figs. 31 and 32). In contrast, no obvious change of events can be found. Therefore, it indicates that these changes were caused by the interaction of light and Au nanoparticles.

Previous studies have proved that light excitation of the SPR of the Au nanoparticles could generate hot electrons and holes via Landau damping[3,6,44,45]. The temperature raised by light irradiation (52 mW cm$^{-2}$, $\lambda_{ex} = 500$) around the Au nanospheres in solution is negligible (less than 1 °C)[17,46]. Thus, the energetic electrons accumulated on the Au nanoparticle while the holes were consumed by reacting with adsorbed molecules and even $Au^0$, which increased the net charge of the Au nanoparticles for easier capture through nanopore[6,44,46]. The thickness of counter-ion cloud on more negative charged Au nanoparticles also increased, leading to a rise in $I_b$. Since the clean surface of Au nanoparticles is exposed through releasing oxidized adsorbed molecules and etching of $Au^0$, the trend of interparticle coalescences should increase. To further demonstrate this mechanism, $I_2$ was replaced by $HAuCl_4$ in the reaction system. High concentration of $AuCl_4^-$ can oxidize the $Au^0$ (Supplementary Fig. 33a) and remove the adsorbed surfactant. TEM result showed that the Au nanoparticles indeed aggregated together, however, with irregular shapes instead of Au nanoplates (Supplementary Fig. 33b). In this case, $I_2$ species should exert a critical influence and cooperate with hot holes for the formation of Au nanoplates.

To prove that the aggregation of Au nanoparticles can make nanoplate, it is necessary to show that the aggregation and coalescence process can produce planar twinned Au nanostructures, which can develop to Au nanoplates. As shown in the Fig. 4a and Supplementary Fig. 34, we traced several linear aggregates using TEM before and after 30 min reaction in the real condition, and found that they coalesced into quasi-spherical nanoparticles. For Supplementary Fig. 34b, the insert HRTEM image clearly show that the linear aggregate changed to a planar twinned nanostructure, because the twin plan (highlighted by the white dash line) is just vertical to the TEM chip. It is not always easy to identify all of these twinned structures with different postures by TEM. Thus, we further grow these nanoparticles with previous reported method (details shown in the figure legend), by which the planar twinned Au nanostructures will develop to Au nanoplates and other twinned Au nanostructures (mainly penta twinned structure) will grow to nanospheres (Fig. 4a and Supplementary Fig. 34)[4]. Indeed, we observed the formation of Au nanoplates. It is clearly demonstrated that planar twinned Au nanostructures can form through nanoparticle-coalescence process.

It has been reported recently that iodide could stabilize hot holes and enable the edge-selective oxidative etching of $Au^0$ sites with lower coordination numbers[6]. Thus, the surfactant on the edge (twin boundary) of Au nanoplate would become easier to be removed, which promoted the nanoparticle coalescence preferentially along the perimeter of the nanoplate. The hot hole-controlled site-selective coalescence process has also been directly observed by TEM tracking of the intermediate structure immobilized on SiNx window (Fig. 4b). It is found that the smaller Au nanosphere on the edge of nanoplate slowly merges into the nanoplate as the reaction proceeds. Coincidently, there is also an Au nanoparticle on the top (or bottom) of the Au nanoplate. However, no obvious change can be observed for it. In order to provide atomistic details on the coalescence process, we performed explicit solvent atomistic molecular dynamics simulations of two Au nanoparticles with different sizes and shapes

(Fig. 4c, Supplementary Fig. 35a, and Videos 2 and 3). The results showed that a larger Au nanoplate would merge a small Au nanosphere on the edge for growth, no matter the nanosphere owns planar twinned or penta twinned structure. However, for the coalescence of two Au nanospheres with similar size, a multiple Au twinned structure will survive (Supplementary Fig. 35b and Video 4). Taken together, a clear picture is established for describing the plasmon-driven conversion from Au nanospheres to Au nanoplates, which is a controllable nanoparticle-coalescence dominated growth process under the cooperation between light irradiation and iodide.

In summary, we conclusively demonstrated that the plasmon-driven anisotropic growth of Au nanoplates can be achieved by nanoparticle-coalescence growth pathway. Nanopore sensing was used as a compatible technology in combination with TEM tracking characterizations and molecular dynamics simulations to exactly illustrate the growth process. Iodide cooperated with the light prompts the hot hole-controlled edge-selective aggregation and coalescence of the Au nanoparticles to Au planar twinned nanostructures, resulting in the growth of Au nanoplates in size. This work conclusively proved that plasmon-driven nanoparticle-coalescence growth mode exists and could generate well-defined anisotropic metal nanostructures. Furthermore, our work suggested that the nanopore sensing technique could be of general use for studying the growth process of nanostructures.

## Methods

**Synthesis of Au nanospheres**. Au nanospheres were prepared using a previously reported method[4,47]. Typically, 1 ml of 10 mM $HAuCl_4 \cdot 3H_2O$ and 1 ml of 10 mM sodium citrate were added to 37 ml of ultrapure water in a 50 ml beaker under magnetic stirring. Then 1 ml of 100 mM $NaBH_4$ (freshly prepared with ice-cold water) was rapidly injected into the solution. The solution turned garnet red, which indicated the formation of Au nanospheres. The resulting Au nanospheres solution was aged for more than 12 h in the dark before use. Finally, the solution of Au nanospheres was diluted with ultrapure water to ensure that the peak intensity at 512 nm was about 0.24.

**Photoinduced conversion of Au nanospheres to nanoplates**. The photo-conversion of Au nanospheres to nanoplates was carried out in a 10 ml glass vial. 170 μl of 0.50 mM iodine ($I_2$) solution ($I_2$ solution needs to be kept for more than 12 h after preparation, but used within 2 weeks) was quickly added to 5 ml of the Au nanospheres solution with gentle shaking. This solution was illuminated for 540 min by a light source of xenon lamp equipped with a 500 ± 10 nm bandpass filter under an incident power of 52 mW cm$^{-2}$. The distance between the vial and the light output window was 15 cm. The dark reaction was performed by wrapping the glass vial with aluminum foil. For wavelength-dependent photoconversion experiments, the solution was illuminated with 400 ± 10 nm, 600 ± 10 nm, and 700 ± 10 nm band-pass filters. All of the photoconversion processes were monitored by UV–Vis spectroscopy by sampling a 200 μl reaction mixture at set intervals.

The synthesized nanoplates were collected by centrifugation at 158 × $g$ for 5 min, and then dispersed in water. This process was repeated two times to remove excess surfactant from the nanostructure surface, before characterizing the sample by SEM and TEM. For intermediate species in the reaction process, 5 μl of the reaction mixture was dropped directly on the copper grid and dried in ambient air. The samples were rinsed with a large amount of ultrapure water and then dried again for TEM observation.

**Nanostructure evolution tracking experiments**. 5 μl of Au nanoparticles or other samples after a certain time of reaction were immobilized on the SiNx window by drop-casting onto the substrate, dried, and cleaned as described above for TEM observation and a location recording. The SiNx window was then immersed in the conversion solution for studying the etching effect of $I_2$ or the transformation process of nanostructure in single particle level.

**Synthesis of 13 nm Au nanospheres**. Citrate-stabilized Au nanospheres (~13 nm) were synthesized following Frens' method as reported previously[48]. Briefly, 150 ml of 0.01 wt% $HAuCl_4 \cdot 3H_2O$ was brought to rolling boil with vigorous stirring. Next, 4.50 ml of 1 wt% sodium citrate was added quickly. The solution was stirred until turned deep red suspension. The resulting solution was stored at room temperature and away from light. The size of Au nanoparticles was 13.50 ± 1.28 nm (Supplementary Fig. 21).

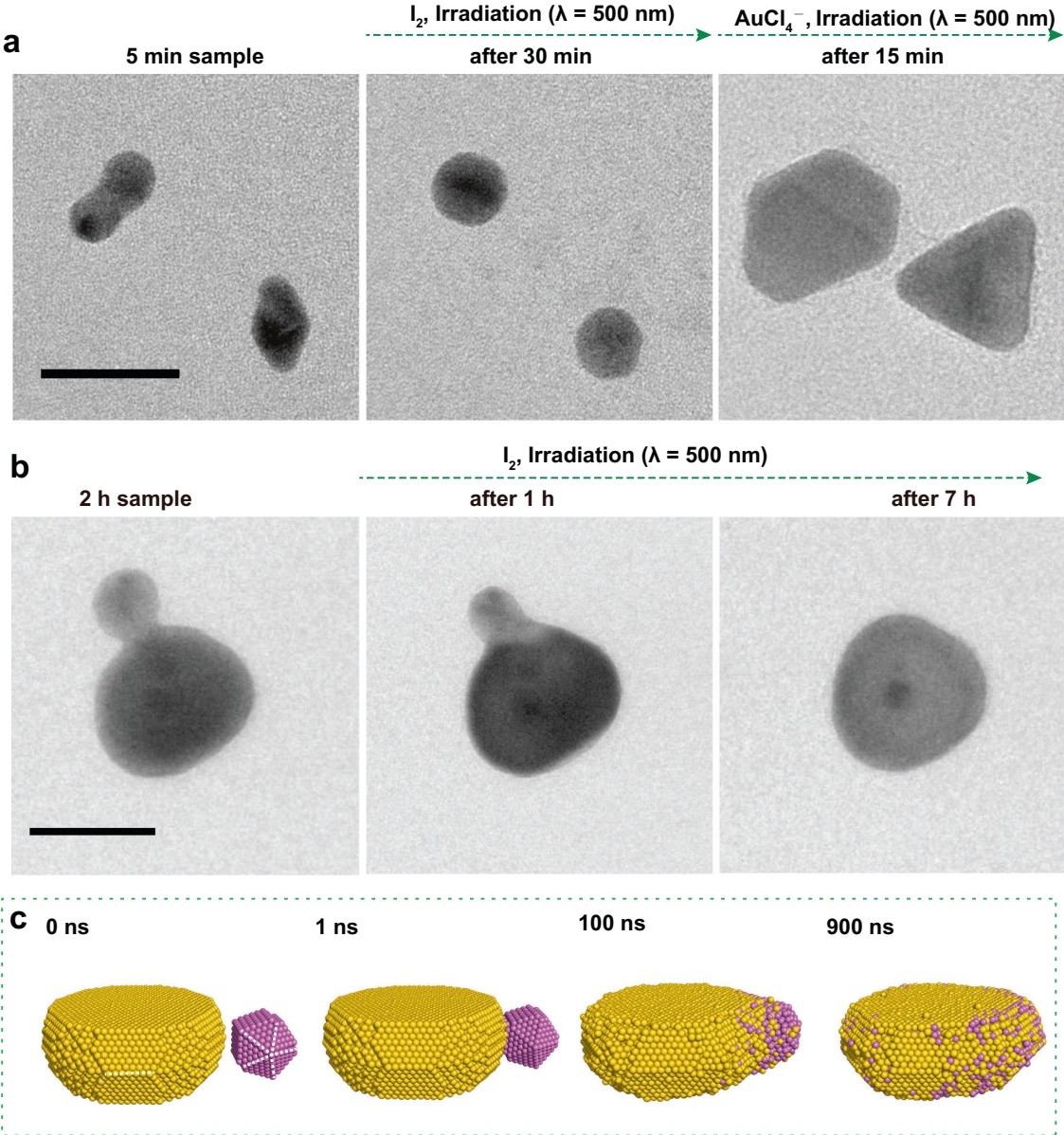

**Fig. 4 Evolution of nanoparticles coalescence process by TEM tracking and MD simulations. a** TEM images of the tracking of nanoparticle aggregates conversion process. The first reaction is to put the TEM chip in the real conversion solution for 30 min with 500 nm light irradiation. The second reaction is to further put the TEM chip in a solution containing 4.84 ml ultrapure $H_2O$, 0.50 ml PVP (K30, 5 mg/ml), 0.50 ml $CH_3OH$, and 0.16 ml $HAuCl_4$ (25 mM) for 15 min under 500 nm light irradiation. The scale bar indicates 20 nm. **b** TEM images of nanoparticle-nanoplate coalescence intermediate structure to nanoplate conversion process. The scale bar indicates 50 nm. **c** Snapshots of the coalescence trajectory of a larger Au nanoplate (gold) and penta twinned structure (pink).

**Synthesis of 50 nm $SiO_2$ nanospheres**. The 50 nm $SiO_2$ nanospheres were prepared as reported by Shi and co-workers[49]. 2.00 g CTAC and 0.06 g TEA were dissolved in 20 ml of ultrapure water at 95 °C under magnetic stirring for 1 h. 1.50 ml of TEOS was added dropwise to the solution and this mixture continued to stir for another 1 h. The products were collected by centrifugation and washed for several times with ethanol.

**Nanopore fabrication and characterization**. Nanopipettes pulled on a laser pipet puller (P2000, Sutter Instruments) using quartz capillaries with filament (QF100-50-7.5 and QF100-70-7.5; Sutter Instrument Co). All quartz capillaries were thoroughly cleaned by immersing in freshly prepared piranha solution (3:1 98% $H_2SO_4$/30% $H_2O_2$) at 80 °C for ~3 h to remove organic impurities. Caution! *Piranha* is highly reactive and corrosive, and can become extremely hot when prepared: handle with care and use appropriate personal protection equipment. After extensive washing using ultrapure water, the capillaries were dried at 60 °C in an oven. The parameters of different diameter nanopores used in the experiment are summarized in Table S1. Please note that the parameters for nanopipettes fabrication are instrument specific and can be different for each individual pipette

puller. The tip of the nanopipette (~1 cm) was cut off and fixed vertically on the side of the aluminum table. The nanopipette was pretreated by sputtering with Pt for 20 s and then characterized by SEM for pore size.

**Nanopore experiment and data analysis**. The as-prepared nanopores were fixed on a homemade PMMA chip. 50 mM $KNO_3$ solution was filled into the nanopipette (*trans* chamber). The sample solution was diluted using an equal volume of 10 mM $KNO_3$ solution and 500 μl test solution was added into the external reservoir of the nanopipette (*cis* chamber). Ag/AgCl electrodes were placed at two sides of the nanopore and different bias voltages were applied during the nanoparticles translocation experiments. The effect of illumination on nanoparticles translocation experiments was carried out in same solution and nanopipette, with all conditions identical except for illumination. All the electrical signals were detected with a 10 kHz four-pole Bessel low-pass filter at a 100 kHz sampling frequency. The entire experimental setup was placed inside a Faraday cage to shield it from electromagnetic interference. The current signals analysis was performed by using the Transalyzer Matlab package[50]. Different types of peaks were extracted

and processed by custom peak analysis batch code based on Perl. The statistical analysis was performed using the SPSS 24.0 software.

**Molecular dynamics simulations of Au nanoparticle coalescence**. To simulate the Au nanoparticle coalescence process, we put two Au nanoparticles whose initial models were prepared by the Nanomaterial Modeler module in CHARMMGUI webserver[51] and VMD software v1.93[52] with manual operations. The structure, topology, and parameters of metals are based on the INTERFACE FF force field[51–53]. To test the dependence of the coalescence mechanism on the nanoparticle topology, we built a series of models in different sizes and shapes (see Supplementary Videos 2–4).

One microsecond simulation was performed for each model. To accelerate the coalescence process, the temperature was kept constant at 900 K using the Nose-Hoover thermostat. Neighbor searching was performed every 20 steps. The PME algorithm was used for electrostatic interactions with a cut-off of 1.20 nm. A reciprocal grid of 80 × 80 × 80 cells was used with 4th order B-spline interpolation. A single cut-off of 1.20 nm was used for Van der Waals interactions. MD simulations were performed using Gromacs 2019.5[54]. The movies were prepared by PyMol2.5 (The PyMOL Molecular Graphics System, Version 2.0 Schrödinger, LLC.).

## Data availability

The datasets used for machine learning are available at https://figshare.com/articles/dataset/Data-Envents/19246224. All data generated during and/or analyzed during the current study are available from the corresponding author on request.

## Code availability

The code for peak analysis of nanopore events and machine learning used in this study are available at https://figshare.com/articles/dataset/Code-Peak-ML/19245825.

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

## Acknowledgements

The authors thank Pengyuan Qi and Junyu Zhang for machine learning coding, and Jiajia Guo for statistical analysis. We are also grateful for the financial support from National Natural Science Foundation of China (No. 21974103), the start-up funds of Wuhan University, and the start-up funds of Zhejiang University.

## Author contributions

B.H. and Y.Z. conceived the project and designed the experiments. B.H. and L.M. synthesized the materials. B.H., L.M., and J.L. performed SEM and TEM characterization. L.M. performed UV–Vis characterization. B.H., L.M., and Z.X. performed nanopore experiments. Y.W. performed molecular dynamics simulations and wrote the peak analysis batch code. B.H., L.M., J.C., and Y.Z. analyzed the data and prepared the manuscript with contributions from all authors. Y.Z. supervised the project.

## Competing interests

The authors declare no competing interests.
