## [Peer Review File · Nature Communications]

REVIEWER COMMENTS

Reviewer #1 (Remarks to the Author):

The authors report a method for the iodine-assisted, plasmon-driven synthesis of Au nanoplates from Au spheres. To understand the reaction mechanism, the authors use a nanopore sensing characterization method to identify aggregation events. This approach is very interesting and is potentially a powerful tool for uncovering coalescence-mediated nanoparticle growth mechanisms. However, in the current work it is unclear that this aggregation directly leads to nanoplate formation as proposed. In addition, the final proposed growth mechanism is somewhat inconsistent with the data discussed in the body of the paper. As a result, I recommend reconsideration after major revisions to address the following concerns:

1. The presence of salt can destabilize citrate-capped gold nanoparticles by screening the surface charge and can lead to aggregation. The authors show in Figure S12 that 5 nm Au spheres are unstable above 10 mM KNO₃ while 13 nm Au spheres are unstable above 30 mM KNO₃. However, when the authors did their control nanopore experiments to show that the salt conditions of the nanopore experiment don't themselves induce aggregation, they chose to use 13 nm Au spheres for the control instead of 5 nm Au spheres. 13 nm Au spheres are significantly more stable towards aggregation than 5 nm spheres, even in the absence of salt, and therefore this control is not sufficient to conclusively show that the high salt concentrations at the interface of the nanopore do not induce aggregation as an artifact (that would not normally be present during nanoplate growth).
2. While the authors' results do show evidence of aggregation, it is not clear how this aggregation leads to nanoplate growth. For example, how would the large linear aggregates shown in Figure S20 rearrange to form a plate shape? It is well-known that 5 nm Au spheres aggregate and coalesce into larger particles (this is why the 5 nm seeds used for seed-mediated syntheses must be made fresh daily), and therefore it would be helpful to see more evidence of how this aggregation directly correlates with nanoplate formation.
3. The authors show examples of a single particle merging into a larger plate structure as well as long chains of aggregated spheres. Some analysis of the reaction order might provide evidence into whether the process of plate formation involves stepwise addition of smaller spheres to larger particles or the concerted merging of many particles at once. The reaction order for plasmon-mediated Ag nanoprism formation has previously been presented as one piece of evidence against the oriented attachment hypothesis for that particular system (J. Am. Chem. Soc. 2008, 130, 9500).

4. On page 16, the authors state that at the beginning of the nanoplate growth reaction “I₂ induces the formation of larger Au planar twinned nanoplates through a fast redox reaction.” This is then proposed to be followed by coalescence. However, on page 7, the authors also claim that deposition of Au through re-reduction is unlikely because NMR shows no evidence of a change in the citrate concentration. Do the authors have evidence for this first stage of growth where nanoplates form through a redox process? For example, does adding Au ions to the reaction solution change the growth kinetics at this stage?

In addition, how does this two-part mechanism fit with the authors’ observations of coalescence of sphere-shaped particles (as opposed to just the coalescence of a plate and a small sphere?) The overall proposed mechanism on page 16 seems somewhat inconsistent with the body of the paper.

5. SiO₂ nanoparticles are used as a control to represent the behavior of particles that do not absorb 500 nm light. However, these particles are likely to have different surface functionalization from the Au spheres as well, and therefore may behave differently for that reason also. (The synthesis and/or surface capping of the SiO₂ spheres is not described, so it is not clear what is on their surface.)

6. In the abstract, the authors state that “the mechanism based on plasmon-driven hot-electron-reduction dominated anisotropic growth...is widely accepted and developed.” This is not necessarily true for Ag nanoparticles, as at least one popular mechanism involves the hot-hole-mediated oxidation of citrate, followed by the reduction of Ag⁺ by the thermalized electrons that result from this oxidation (J. Phys. Chem. C 2007, 111, 8942).

7. In Figure 1, it would be helpful for part (g) to be scaled to the same y-axis as (f) for ease of comparison. The spectra in (g) are somewhat squashed vertically.

Reviewer #2 (Remarks to the Author):

Huang et al. reported the translocation events of the intermediate species of nanoparticles during photosynthesis which the morphology conversion was considered as a hot hole-controlled coalescence-dominated growth process. It would be of great importance at single-entity level that the intermediate states could be monitored with high resolutions. However, the methodology used

in this work is one of the conventional nanopore detections without further improvements. The insights are not very convincing based on their limited correlation between noisy current recording and limited TEM images, which hinders the impact of the work as well as broader interests. To this end, I would suggest a major revision or transfer to Scientific Report.

1. The reaction condition of photo-induced conversion in Figure 1 is not the same as the rest of nanopore experiments, lacking of 50 mM KNO₃ as the electrolyte, which has to be carefully addressed.
2. Although gold nanoparticles do not aggregate at 5 mM or 50 mM KNO₃, it will be important to perform dynamic light scattering (DLS) measurements to ensure no aggregation occurred during your storage or upon previous operations.
3. Please provide the diameters, IV curves and baseline RMS noises of all the nanopores used in Figure 2 and 3 as well as in rest of this work.
4. Could you explain the reasons why you used ~20 nm, 1 V for Au nanospheres but ~120 nm, 300 mV for Au nanoplates in Figure 2? Which condition do you use in Figure 3? Could you provide the current at 0h? What are the baseline currents for 0h, 1h, 3h, 5h and 9h in Figure S26?
5. How does the light shine onto your system? Is it focused at the nanopore or passing through the entire trans/cis side of the flowcell? For the latter case, you then measured the diameter evolution more than the light-induced intermediate species, didn't you?
6. It would be critical to understand the min and max hydrodynamic diameters of gold nanoparticles that a nanopore with a certain diameter can cover. Since 9h measurements were performed on 120 nm pores, the cross-section $(5 \text{ nm}/120 \text{ nm})^2$ should lead to negligible deviations from my point of view. Could you identify the range of particle size your nanopores can detect above your noise level?
7. The reported mean dwell time (Venta et al. Nano Lett. 2014, 14, 9, 5358–5364) of single gold particle translocations is less than 0.1 ms. Could you please make a full description and explains how you make your device work since I am a bit lost in this part?

8. No scales and units in Figure S23. Figure S26 is displayed without detailed experiment conditions. Please make sure the entire piece of work is consistent and well-organized.

9. TEM observations are fantastic. Could you do more statistics based on a decent amount of TEM images to correlate with translocation events?

Reviewer #3 (Remarks to the Author):

The authors present a nice study of their work in supporting a coalescence mechanism as the dominant mechanism of shape transformation of plasmonic nanoparticles. There is a lot of good information within this article to support the proposed mechanism.

1) The authors state that "the mechanism governing

the morphology conversion is still elusive, which limits its broader extension". I do not really agree with this. There are two mechanisms at play as proposed in references like #2 and supported by that reference (optical data) and many of the other references used herein. The mechanism is not elusive it has two different components, that are at play in different proportional amounts like with oxygen present, and Ag as the metal the induction period is dominated by Ostwald ripening type processes but reference 2 clearly shows optical signatures for aggregates later in the process when coalescence dominates. The study presented here is not surprisingly dominated by coalescence since the metal is gold which is more easily maintained in a reduced state.

The authors do present a lot of very good data to support this mechanism though.

2) One technical issue I have is with the absorbance spectra. They are repeatedly referred to as UV-vis spectra, which is common in literature but absolutely wrong. The y-axis in all of these plots is labelled "Intensity (a.u.)". I think the authors mean Absorbance, which is not a.u. but a unitless quantity and should be written as such. Absorbance is not an intensity measurement, actually it is a lack of intensity.

3) Further to point #2 the absorbance spectra throughout the Supplemental information have absorbances below zero which is not possible these need to be explained at the very least, if there is a background subtraction being used, and probably fixed actually since this data makes no sense.

Response to Reviewers

We appreciate all the reviewers for their valuable and helpful comments in improving our manuscript. The detailed and thoughtful responses have strengthened this manuscript. All the responses are shown in blue. In the revised manuscript and supplementary information, all the revised and new data have been highlighted in blue.

Reviewer #1 (Remarks to the Author):

The authors report a method for the iodine-assisted, plasmon-driven synthesis of Au nanoplates from Au spheres. To understand the reaction mechanism, the authors use a nanopore sensing characterization method to identify aggregation events. This approach is very interesting and is potentially a powerful tool for uncovering coalescence-mediated nanoparticle growth mechanisms. However, in the current work it is unclear that this aggregation directly leads to nanoplate formation as proposed.

In addition, the final proposed growth mechanism is somewhat inconsistent with the data discussed in the body of the paper. As a result, I recommend reconsideration after major revisions to address the following concerns:

Authors' response: We are very grateful for the valuable comments. With the help of the comments, we have collected extra evidences to make our proposed mechanism more convincing. We also further modified the mechanism, and the description of photo-driven morphology conversion of Au nanoparticles should be clearer.

1. The presence of salt can destabilize citrate-capped gold nanoparticles by screening the surface charge and can lead to aggregation. The authors show in Figure S12 that 5 nm Au spheres are unstable above 10 mM KNO_3 while 13 nm Au spheres are unstable above 30 mM KNO_3 . However, when the authors did their control nanopore experiments to show that the salt conditions of the nanopore experiment don't themselves induce aggregation, they chose to use 13 nm Au spheres for the control instead of 5 nm Au spheres. 13 nm Au spheres are significantly more stable towards aggregation than 5 nm spheres, even in the absence of salt, and therefore this control is not sufficient to conclusively show that the high salt concentrations at the interface of the nanopore do not induce aggregation as an artifact (that would not normally be present during nanoplate growth).

Authors' response: We are very grateful for the valuable comments. Because well-defined monodispersed 13 nm Au nanoparticles were easily obtained, they were used for control experiment to demonstrate the aggregation is not artifact caused by the detection method. The control experiment with 13 nm is also very meaningful. The TEM (Supplementary Figure 9) results show that the size of Au nanoparticles is already close to or more than 13 nm after 30 min of growth reaction (Fig. R1). Therefore, nanopore analysis using 13 nm Au nanoparticles as the control experiment can exclude the artifact during conversion process (over 30 min).

To make a more convincing control experiment for the smaller size Au nanoparticles, we tried to obtain monodisperse Au seed nanoparticles by centrifugation to get rid of most of

large aggregations. As shown in Fig. R2, Au nanoparticles with small size and much better monodispersity were obtained. The relative translocation events are mainly single peak events (more than 90%), indicating that the salt concentration gradient does not induce aggregation as an artifact.

Fig. R1. The TEM image (left) and histogram (right) of the size distribution of nanostructure from 30 min sample.

Fig. R2. Analysis translocation events of Au nanospheres (after centrifugation). (a) Concatenate events from Au nanospheres (10000 rpm, 15 min) translocation test using a ~ 20 nm quartz nanopore at a 1 V bias voltage. (b) Scatter plots of t_d vs. I_b . (c) Pie chart of four types of peaks in a total of 1408 translocation events. The percentage of single, double, triple, and multiple peaks were 91.97%, 3.76%, 1.56% and 2.71%, respectively. (d) TEM image of 5 nm Au nanospheres from supernatant after centrifugation.

2. While the authors' results do show evidence of aggregation, it is not clear how this aggregation leads to nanoplate growth. For example, how would the large linear aggregates shown in Figure S 20 rearrange to form a plate shape? It is well-known that 5 nm Au spheres aggregate and coalesce into larger particles (this is why the 5 nm seeds

used for seed-mediated syntheses must be made fresh daily), and therefore it would be helpful to see more evidence of how this aggregation directly correlates with nanoplate formation.

Authors' response: We are very grateful for the valuable comments. Taking advantage of TEM tracking and nanopore analysis, it is demonstrated that aggregations happen during the transformation of Au nanospheres into nanoplates in our system.

To prove that the aggregation of Au nanoparticles can make nanoplate, it is necessary to show that the aggregation and coalescence process can produce planar twinned Au nanostructures, which can develop to Au nanoplates. Here, we utilize the TEM tracking strategy to get new evidences. As shown in the Fig. R3, we traced several linear aggregates using TEM before and after 30 min reaction in the real condition, and found that the aggregate coalesced into one nanostructure. For Fig. R3c, the insert HRTEM image clearly show that the linear aggregate changed to a planar twinned nanostructure, because the twin plan (highlighted by the white dash line) is just vertical to the TEM chip. It is not always easy to identify all of these twinned structures with different postures by TEM. Thus, we further grow these nanoparticles with previous reported method (details shown in the figure legend), by which the planar twinned Au seeds will develop to Au nanoplates and other twinned Au seeds (mainly penta twinned structure) will grow to nanospheres (*Nat. Mater.*, **2016**, 15(8): 889-895) (Fig. R3, the third row). Indeed, we observed the formation of Au nanoplates. It is clearly demonstrated that planar twinned Au nanostructures can form through nanoparticle-coalescence process. The stronger absorption for iodide species on Au (111) facet should play very important role for this conversion, since common aggregations always produce complicated twinned nanostructures, such as penta twinned nanoparticles (Supplementary Figure 33; the MD calculation, Supplementary Figure 35; *Science*, **2020**, 367(6473): 40-45). The planar twinned Au nanostructures can further grow to Au nanoplates.

Fig. R3. TEM images of the tracking of nanoparticle aggregates conversion process. The first reaction is to put the TEM chip in the real conversion solution for 30 min with 500 nm light irradiation (the second row). The second reaction is to further put the TEM chip in a solution containing 4.84 ml ultrapure H₂O, 0.50 ml PVP (K30, 5 mg/ml), 0.50 ml CH₃OH, and 0.16 ml HAuCl₄ (25 mM) for 15 min under 500 nm light irradiation (the third row). All scale bars indicate 20 nm.

3. The authors show examples of a single particle merging into a larger plate structure as well as long chains of aggregated spheres. Some analysis of the reaction order might provide evidence into whether the process of plate formation involves stepwise addition of smaller spheres to larger particles or the concerted merging of many particles at once. The reaction order for plasmon-mediated Ag nanoprism formation has previously been presented as one piece of evidence against the oriented attachment hypothesis for that particular system (*J. Am. Chem. Soc.*, **2008**, 130, 9500).

Authors' response: We are very grateful for the valuable comments. According to our observations, the linear aggregation of Au nanospheres can merge together fast to form planar twinned nanostructure in the reaction condition, as shown in the Fig. R3. In the initial stage, we have not observed the intermediate structures with one nanoplate merging with

many nanoparticles at one time. Thus, the stepwise aggregation may exist as the reviewer mentioned. It is proposed that the Au nanoplate mainly form with coalescence of Au nanoparticles one by one. However, when the Au nanoplate grow in size. There are opportunities for the coalescence of more Au nanoparticles on one Au nanoplate (Fig. R4).

Based on the evidences from TEM tracking and nanopore analysis, the plasmon-driven conversion of Au nanospheres to nanoplates indeed follows a different mechanism towards most reports for the silver counterpart, which is proposed mainly following the oxidation and re-reduction process.

Fig. R4. TEM images of the Au nanoparticles on nanoplate. All scale bars represent 20 nm.

4. On page 16, the authors state that at the beginning of the nanoplate growth reaction “ I_2 induces the formation of larger Au planar twinned nanoplates through a fast redox reaction.” This is then proposed to be followed by coalescence. However, on page 7, the authors also claim that deposition of Au through re-reduction is unlikely because NMR shows no evidence of a change in the citrate concentration. Do the authors have evidence for this first stage of growth where nanoplates form through a redox process? For example, does adding Au ions to the reaction solution change the growth kinetics at this stage? In addition, how does this two-part mechanism fit with the authors’ observations of coalescence of sphere-shaped particles (as opposed to just the coalescence of a plate and a small sphere?) The overall proposed mechanism on page 16 seems somewhat inconsistent with the body of the paper.

Authors’ response: We are very grateful for the valuable comments. In the previous manuscript, the growth process is described as two parts: fast redox reaction and slow coalescence-dominated growth. For the coalescence-dominated growth, strong evidences can demonstrate this process. For example, NMR shows no obvious change was found in the concentration of citrate; intermediate aggregates were characterized by TEM tracking and nanopore detection. Since we did not have solid evidence that the nanoparticle-coalescence can produce planar twinned nanostructures then, we proposed the redox-dominated process for the very beginning stage, which is quite similar to the reported mechanism of silver system.

However, in the revised manuscript, according to new evidences, we have got a more comprehensive description for the growth process. It is demonstrated that the aggregation and coalescence can produce planar twinned Au nanostructures. Thus, the nanoparticle-coalescence growth pathway is proposed to dominate for the whole growth process.

At the beginning of the reaction, the adding of I_2 can selectively etch the Au

nanoparticles (Supplementary Figure 6). It is still possible that the released Au ions can be re-reduced and deposited onto more stable Au nanoparticles. The citrate can act as reducing agent, but the consumption amount is too small to be clearly detected by NMR. To prove it, as the reviewer suggested, we added additional HAuCl_4 in the solution. As shown in Fig. R5, the stronger extinction peaks exhibit the growth of the Au nanostructures. However, no obvious dominated SPR peak of larger Au nanoplates appeared for the sample with additional 100 μl HAuCl_4 (10 mM) after 180 min light irradiation. It indicates that the addition of Au ions in the conversion solution is not helpful for the selective generation of Au nanoplates.

We have updated the growth mechanism in the revised manuscript, in which the nanoparticle-coalescence growth dominates in the whole growth process.

Fig. R5. Time-dependent UV-Vis spectra of conversion solution and after adding different volumes of 10 mM HAuCl_4 .

5. SiO_2 nanoparticles are used as a control to represent the behavior of particles that do not absorb 500 nm light. However, these particles are likely to have different surface functionalization from the Au spheres as well, and therefore may behave differently for that reason also. (The synthesis and/or surface capping of the SiO_2 spheres is not described, so it is not clear what is on their surface.)

Authors' response: We are very grateful for the valuable comments. The SiO_2 nanoparticles were prepared as reported method (*J. Am. Chem. Soc.*, **2012**, 134(13): 5722-5725.), which is added in the revised manuscript. The surface of SiO_2 was mainly

capped by CTAC. Because the SiO₂ nanoparticles do not absorb 500 nm light, the purpose of the control experiment with SiO₂ nanoparticles is to demonstrate that the change of translocation events through nanopore is caused by the absorption of light by the Au nanoparticles, not other effects that causing changes to the nanopore or solution. Because the surfactants (such as citrate and CTAC) do not absorb 500 nm light, the plasmon excitation of Au nanoparticle should play the main role as we discussed in the manuscript.

6. In the abstract, the authors state that “the mechanism based on plasmon-driven hot-electron-reduction dominated anisotropic growth...is widely accepted and developed.” This is not necessarily true for Ag nanoparticles, as at least one popular mechanism involves the hot-hole-mediated oxidation of citrate, followed by the reduction of Ag⁺ by the thermalized electrons that result from this oxidation (*J. Phys. Chem. C.*, **2007**, 111, 8942).

Authors' response: We are very grateful for the valuable comments. We have changed these sentences accordingly.

7. In Figure 1, it would be helpful for part (g) to be scaled to the same y-axis as (f) for ease of comparison. The spectra in (g) are somewhat squashed vertically.

Authors' response: We are very grateful for the valuable comments. We have plotted Fig. 1g as suggested by the reviewer.

Reviewer #2 (Remarks to the Author):

Huang et al. reported the translocation events of the intermediate species of nanoparticles during photosynthesis which the morphology conversion was considered as a hot hole-controlled coalescence-dominated growth process. It would be of great importance at single-entity level that the intermediate states could be monitored with high resolutions. However, the methodology used in this work is one of the conventional nanopore detections without further improvements. The insights are not very convincing based on their limited correlation between noisy current recording and limited TEM images, which hinders the impact of the work as well as broader interests. To this end, I would suggest a major revision or transfer to Scientific Report.

Authors' response: We are very grateful for the valuable comments. In this work, we observed the direct plasmon-driven morphology conversion from Au nanospheres to Au nanoplates, which is a very interesting process. To confirm the growth way, we mainly utilized both TEM tracking and nanopore detection to study the intermediate species. The main contribution of nanopore sensing technology is that it plays a very important role in confirming intermediate aggregated states at single-entity level in the solution, while there are significant limitations to confirm it under TEM characterization (even with liquid-cell TEM).

Additionally, as a powerful tool, we also demonstrated that the nanopore sensing could be of general use for studying the growth mechanism of nanostructures. The salt concentration gradient strategy has been used for the research of DNA and proteins, we demonstrated that it is indeed helpful for nanopore analysis of metal nanoparticles, improving nanoparticle capture rates and translation time resolution. We present here the strategy of determining the shape and coalescence of nanoparticle in solution using nanopore sensing.

In the revised manuscript, we strengthened the studies of the capability of nanopore detection and TEM tracking for the nanoparticle conversion. We believe that this revised manuscript will be of great interest to researchers working on better understanding plasmon-mediated syntheses and studying the nanostructures shape evolution.

1. The reaction condition of photo-induced conversion in Figure 1 is not the same as the rest of nanopore experiments, lacking of 50 mM KNO_3 as the electrolyte, which has to be carefully addressed.

Authors' response: We are very grateful for the valuable comments. For the nanopore characterization, certain amount of growth solution is taken out and KNO_3 electrolyte was added to support the electrochemical analysis. It is demonstrated that the present of 5mM KNO_3 will not cause aggregation of Au nanoparticles, which is good for capturing the intermediate species by nanopore sensing. Additionally, no light irradiation was supplied during the morphology characterizations with nanopore. With nanopore sensing, we did not directly monitor the evolution of nanostructures, but captured the intermediates for different reaction time.

2. Although gold nanoparticles do not aggregate at 5 mM or 50 mM KNO_3 , it will be important to perform dynamic light scattering (DLS) measurements to ensure no aggregation occurred during your storage or upon previous operations.

Authors' response: Thank you for your valuable suggestions. We have supplemented time-dependent DLS of initial Au seeds (Fig. R6). The dominant size distribution shows no significant aggregation of seeds in 5 mM KNO_3 solution during one hour. It suggests that 5 mM KNO_3 does not lead to particle aggregation during nanopore testing. Combined with the control experiments of nanopore characterizations with monodispersed ~5 nm and ~13 nm Au nanoparticles (Supplementary Figure 21 and Figure 22), we further confirm the advantages of the nanopore sensing strategy without artifact.

Fig. R6. Time-dependent DLS of ~5 nm nanospheres

3. Please provide the diameters, IV curves and baseline RMS noises of all the nanopores used in Figure 2 and 3 as well as in rest of this work.

Authors' response: Thank you for your constructive suggestions. The information about the nanopores used in this manuscript is shown in Supplementary Figure 13 (diameters and I-V curves). The baseline RMS noises of different nanopores are now shown in the I-t trace.

Fig. R7. Nanopore size characterization. Current-voltage curves and SEM images of nanopipettes with different nanopore sizes used in the measurements. I-V curves obtained in 50 mM/5 mM trans/cis KNO_3 solution. All scale bars in all SEM images represent 100 nm.

4. Could you explain the reasons why you used ~20 nm, 1 V for Au nanospheres but ~120 nm, 300 mV for Au nanoplates in Figure 2?

Authors' response: We are very grateful for the valuable comments. The nanopore size was chosen by the size distribution of nanostructures from TEM analysis to ensure that it can cover the range of the particle size. The voltage was chosen to ensure a high event rate and to allow the nanopore to work for a long time without blocking. For example, as shown in the Fig. R8, when the applied voltage is 400 mV during the tests of Au nanoplates, obvious blockage occurs after ~200 s. Therefore, it is very important to select appropriate conditions to obtain nanopore signals with high signal-to-noise ratios and high event rates.

Fig. R8. I-t trace of a 9 h sample obtained from a 120 nm nanopore at a bias voltage of 400 mV.

Which condition do you use in Figure 3? Could you provide the current at 0 h? What are the baseline currents for 0 h, 1 h, 3 h, 5 h and 9 h in Figure S 26?

Authors' response: All experimental details are now shown in caption of Supplementary Figure 27. We have provided the current of the initial Au nanoparticles as 0 h. For the point in time when I_2 was added, it is hard to get good translocation signals of the nanoparticles. The system needs several minutes to reach equilibrium. All the baseline currents are now shown in I-t trace.

Fig. R9. The Current-time trace of 0 h sample.

5. How does the light shine onto your system? Is it focused at the nanopore or passing through the entire trans/cis side of the flowcell? For the latter case, you then measured the diameter evolution more than the light-induced intermediate species, didn't you?

Authors' response: We are very grateful for the valuable comments. During nanopore analysis of the shape of the Au nanoparticles, no light shined on the system. Thus, intermediate species were captured for detection. Only for studying the effect of light irradiation on Au nanoparticles, the light was used and passed through the entire solution.

6. It would be critical to understand the min and max hydrodynamic diameters of gold nanoparticles that a nanopore with a certain diameter can cover. Since 9h measurements were performed on 120 nm pores, the cross-section $(5 \text{ nm}/120 \text{ nm})^2$ should lead to negligible deviations from my point of view. Could you identify the range of particle size your nanopores can detect above your noise level?

Authors' response: We are very grateful for the valuable comments. We totally agree with the reviewer that it is important whether the nanopore we choose can cover the nanoparticles in the sample. The sample with the longest reaction time used for nanopore characterization is from 9h reaction. As shown in Fig. R10, the size distribution of nanoplates in the 9 h sample is around 100 nm, while the nanoparticles in the sample are widely distributed with a minimum of about 35 nm (not 5 nm seeds). We further verified the translocation of 35 nm nanoparticles using a 120 nm nanopore with a voltage of 300 mV. Distinct nanoparticle translocation events can be detected. Therefore, we believe that it could match the nanoparticle distribution range of the 9 h sample using a 120 nm nanopore.

Fig. R10. TEM images (a) and histogram (b) of the size distribution of nanostructures from 9 h sample. Pie chart of nanoparticles and nanoplates percentage content from TEM and nanopore (c). The TEM results contain 319 samples and the nanopore results contain 1198 events. I-t trace and corresponding TEM image and size distribution of 35 nm Au nanoparticles (d).

7. The reported mean dwell time (Venta et al. *Nano Lett.* **2014**, 14, 9, 5358–5364) of single gold particle translocations is less than 0.1 ms. Could you please make a full description and explains how you make your device work since I am a bit lost in this part?

Authors' response: We are very grateful for the valuable comments. We suppose that this difference may be caused by the method of salt gradient and the conical nanopipette nanopore used in this work, while a 40 nm-thick SiNx nanopore was used in the 1 mM to 100 mM KCl solution for both cis and trans sides in the mentioned report (*Nano Lett.* **2014**, 14, 9, 5358–5364). As we described in Supplementary Figure 14 the conical nanopore could introduce a reverse flow by a salt concentration gradient (induced reverse

electroosmotic flow, IREOF). When the Au nanoparticles or nanoplates are added into the low salt concentration, it is easier to be captured by the inlet flow (e.g., IREOF) near the nanopore (*Anal. Chem.*, **2016**, 88, 9251–9258). Then, the Au nanoparticle or nanoplate translocated to the high concentration, it would be retarded by the stronger electroosmosis force (the direction of the electroosmosis force is opposite to the sample movement), resulting in taking the longer time pass through the nanopore (*Biophys. J.*, **2013**, 105, 776–782; *Nat. Nanotechnol.*, **2010**, 5, 160–165). Therefore, the dwell time of Au nanoparticles translations is longer in this work.

Fig. R11. Schematic illustration of a conical nanopore with a salt gradient for nanoparticles translocation.

8. No scales and units in Figure S 23. Figure S 26 is displayed without detailed experiment conditions. Please make sure the entire piece of work is consistent and well-organized.

Authors' response: We are very grateful for the valuable comments. We have revised these figures, and double-checked and revised the manuscript to avoid such problems.

9. TEM observations are fantastic. Could you do more statistics based on a decent amount of TEM images to correlate with translocation events?

Authors' response: We are very grateful for the valuable comments. It would be more informative to combine the statistics of all TEM images with the results of the translocation events. In this work, we mainly utilized the nanopore sensing technique to confirm the aggregation of nanoparticles and calculate the percentage of the Au nanoplates and Au nanospheres for different reaction time. As shown in Fig. R10, for the 9 h sample, the counting of the percentage of Au nanoplates (~88.09%) and Au nanospheres (~11.91%) are close to the statistical analysis from nanopore sensing results (~ 85.98% for nanoplates; ~ 14.02% for nanospheres). For the 5 h sample (as shown in Fig. R12), the counting of the percentage of Au nanoplates (~60.38%) and Au nanospheres (~39.62%) are also close to the statistical analysis from nanopore sensing results (~ 51.83% for nanoplates; ~ 48.17% for nanospheres). However, we encountered problems to do the similar calculations for samples from shorter time reactions, such as 3 h sample (as shown in Fig. R13). It is hard to identify the Au nanoplates from the Au nanospheres when their sizes are close, such as the Au nanostructures marked with red circles in Fig. R13a and 14a. The nanopore analysis

exhibits much better capability to identify the Au nanoplate from the special shape of the translocation events (~41.45% for nanoplates; ~58.55% for nanospheres), and show higher percentage of Au nanoplates compared with the counting result from TEM images (~23.94% for nanoplates; ~76.06% for nanospheres). Obviously, certain amount of Au nanoplates is treated as Au nanospheres in the TEM images. Another challenge is that it is hard to identify the aggregates (e.g., 1 h sample, Fig. R14) from the aggregation artifacts caused by the TEM sample preparation. The subjective judgement for the nanostructures from TEM may cause large deviation for the results. As shown in Fig. R14b, there is a large difference between TEM results and nanopore results in the 1 h sample statistics. Nanoplates only accounted for 9.62% from TEM results. However, the content of nanoplates in nanopore events reached 31.81%. However, for the well-dispersed 13 nm nanoparticles (Fig. R15a) and pure nanoplates (Fig. R10c), the results of TEM statistics and the nanopore translocation events can be correlated well with each other.

Overall, in this system, the statistics analysis from TEM is not suitable to supply correct information for the aggregation states and the calculation of the percentage of Au nanostructures in the early stage. Instead, the nanopore sensing and analysis is very helpful to supply these statistics information. However, the TEM tracking method plays very important role for studying the growth process, as we discussed in the manuscript and responses.

Fig. R12. TEM images (a) of nanostructures from 5 h sample. Pie chart of nanoparticles and nanoplates percentage content from TEM and nanopore (b). The TEM results contain 205 samples and the nanopore results contain 1283 events.

Fig. R13. TEM images (a) of nanostructures from 3 h sample. Pie chart of nanoparticles and nanoplates percentage content from TEM and nanopore (b). The TEM results contain 447 samples and the nanopore results contain 1035 events.

Fig. R14. TEM images (a) of nanostructures from 1 h sample. Pie chart of nanoparticles and nanoplates percentage content from TEM and nanopore (b). The TEM results contain 395 samples and the nanopore results contain 1135 events.

Fig. R15. TEM images (a) of 13 nm nanoparticles. Pie chart of 13 nm nanoparticles in different aggregation state percentage content from TEM and nanopore (b). The TEM results contain 278 samples and the nanopore results contain 2526 events.

Reviewer #3 (Remarks to the Author):

The authors present a nice study of their work in supporting a coalescence mechanism as the dominant mechanism of shape transformation of plasmonic nanoparticles. There is a lot of good information within this article to support the proposed mechanism.

1) The authors state that "the mechanism governing the morphology conversion is still elusive, which limits its broader extension". I do not really agree with this. There are two mechanisms at play as proposed in references like #2 and supported by that reference (optical data) and many of the other references used herein. The mechanism is not elusive it has two different components, that are at play in different proportional amounts like with oxygen present, and Ag as the metal the induction period is dominated by Ostwald ripening type processes but reference 2 clearly shows optical signatures for aggregates later in the process when coalescence dominates.

Authors' response: We are very grateful for the valuable comments. We agree that the sentence "the mechanism governing the morphology conversion is still elusive, which limits its broader extension" was not an accurate description. We have changed it in the revised manuscript.

The study presented here is not surprisingly dominated by coalescence since the metal is gold which is more easily maintained in a reduced state.

The authors do present a lot of very good data to support this mechanism though.

2) One technical issue I have is with the absorbance spectra. They are repeatedly referred to as UV-vis spectra, which is common in literature but absolutely wrong. The y-axis in all of these plots is labelled "Intensity (a.u.)". I think the authors mean Absorbance, which is not a.u. but a unitless quantity and should be written as such. Absorbance is not an intensity measurement, actually it is a lack of intensity.

Authors' response: We are very grateful for the valuable comments. In this article, nanoparticle solutions are usually characterized by measuring their extinction spectra with a UV-vis spectrometer. Although sometimes referred to as absorption spectra, extinction is actually measured as the sum of absorption and scattering of light by the nanoparticles. Absorption dominates for small nanoparticles, and scattering for large ones, but for intermediate sizes (in the range of 40–100 nm), absorption and scattering can be of a similar order of magnitude (Craig F. Bohren and Donald R. Huffman. Absorption and scattering of light by small particles. *John Wiley & Sons*, 2008.). Therefore, we have now updated UV-vis graphs in the revised manuscript using extinction for all nanoparticles and absorbance for solution without a unit quantity.

3) Further to point #2 the absorbance spectra throughout the Supplemental information have absorbances below zero which is not possible these need to be explained at the very least, if there is a background subtraction being used, and probably fixed actually since this data makes no sense.

Authors' response: We are very grateful for the valuable comments. There should be a wrong background subtraction for the absorbance below zero, supplemental Fig. 5a, 7b, and 8a, which were supplemented. We have collected new data and updated them in the revised manuscript. They do not change the findings or interpretation in the text. Thanks very much for pointing out this mistake.

REVIEWER COMMENTS

Reviewer #1 (Remarks to the Author):

The authors have addressed my concerns.

Reviewer #2 (Remarks to the Author):

The authors did a lot of efforts on the improvements of the quality of this work which I sincerely appreciate. However, I become even less convinced at this stage by their fragmentations of nanopore translocation evidences by applying a 5 kHz filter (I guess a low-pass filter) to identify the events. The translocation experiments were performed by transferring some of the reacted solutions out of the bulk into the nanopore device, which is definitely misled in the abstract of the striking phrase “during reaction”. Here are the details which I suggest the authors to re-organize them in-depth. Otherwise, a rejection or transferring to other journals may not be a bad idea.

1. For Q4 R2, leaving all the scatter plots in the Supporting information is definitely not great. I like Figure S27, S28 and S31, however, some of them should be in the main text but not some of the representative events instead. Moreover, these figures are not at the same scale which disappoint me once more.

(1) When you use different sizes of nanopores for the reacted solutions stopped at different time, smaller aggregations will be ignored by larger pore sizes. How could you compare the degree of your aggregations?

(2) Since you have applied a 5 kHz filter (I guess a low-pass filter) to identify the events, then the events faster than 0.2 ms in a lot of your measurements (i.e. Figure S16, S21b, S22d to name a few) cannot be trustable at all. Although the event ratio seems correlated with TEM observation but once you have all the events below 0.2 ms removed, what is going to happen?

(3) Was Figure S31a without light illumination the same as Figure S16? Then what is the pie chart with 500nm illumination going to be?

(4) For 5 nm gold spheres the average translocation time of 0.372 ms was longer than that (0.1 ms) of 13 nm gold spheres in Figure S21b. Is this reasonable?

2. For Q7 R2, the authors did not correlate the references with my question. Anal. Chem., 2016, 88, 9251-9258 and Biophys. J., 2013, 105, 776-782 were simulation works. Following this, what is the predicted average translocation time considering the surface charge density of your gold spheres and the shape of your nanopores? Nat. Nanotechnol., 2010, 5, 160-165 was the experimental work on (highly charged) DNA through SiN nanopores. How would this apply to your case?

Response to Reviewer

We appreciate the reviewer for the valuable and helpful comments in improving our manuscript. The detailed and thoughtful responses have strengthened this manuscript. All the responses are shown in blue. In the revised manuscript and supplementary information, all the revised parts have been highlighted in blue.

Reviewer #2 (Remarks to the Author):

The authors did a lot of efforts on the improvements of the quality of this work which I sincerely appreciate. However, I become even less convinced at this stage by their fragmentations of nanopore translocation evidences by applying a 5 kHz filter (I guess a low-pass filter) to identify the events. The translocation experiments were performed by transferring some of the reacted solutions out of the bulk into the nanopore device, which is definitely misled in the abstract of the striking phrase “during reaction”. Here are the details which I suggest the authors to re-organize them in-depth. Otherwise, a rejection or transferring to other journals may not be a bad idea.

Authors' response: We appreciate the reviewer's comments on the quality of the work. All the electrical signals were detected with a 10 kHz four-pole Bessel low-pass filter (Methods part), and then the events longer than 0.1 ms are reliable. In this system, the dominated events are all longer than 0.1 ms. 5 kHz Gaussian filter was used to smooth obtained current spikes (Figure S15), which will not change the number of captured events. In the abstract, we have removed the term “during reaction”.

In this work we utilized that spherical nanoparticles, nanoplates and their aggregates display different peak shapes for their translocating current blockade through a conical pore. Based on the results from nanopore analysis, TEM tracking experiments, and analysis of the oxidation and reduction of the Au nanostructures, we provide solid evidence that plasmon-driven anisotropic growth of Au nanoplates can be achieved by nanoparticle-coalescence growth pathway.

1. For Q4 R2, leaving all the scatter plots in the Supporting information is definitely not great. I like Figure S27, S28 and S31, however, some of them should be in the main text but not some of the representative events instead. Moreover, these figures are not at the same scale which disappoint me once more.

Authors' response: We are very grateful for the valuable suggestions. We have added the scatter plots of Fig. S31 in a revised Fig. 3 as suggested by the reviewer.

Updated Fig. 3. **Nanopore analysis of the morphological conversion mechanism of Au nanospheres to Au nanoplates.** **a**, The ratio of nanoplates and coalescence in the process of morphology transformation obtained from CNN prediction model and peak analysis code, respectively. 1135, 1035, 1283 and 1198 events were obtained in the 1 h, 3 h, 5 h and 9 h samples, respectively. A representative current trace and event scatter plots of dwell time vs. current blockade of all the experiments are displayed in Supplementary Fig. 27 and 28. **b**, Typical nanopore current–time peak shape for nanoparticle-nanoparticle, nanoparticle-nanoplate and nanoplate-nanoplate coalescence events (left to right). **c**, Current versus time trace of Au nanospheres without (left) and with 500 nm light (right). **d**, Scatter plots for t_d vs. I_b of Au nanospheres translocation without (left) and with 500 nm light (right) from **c**, and along the sides are the corresponding histograms. Measurements were collected using a same ~ 20 nm nanopore at a bias voltage of 1 V. The baseline current is about 310 pA and RMS is 2.18 pA. **e**, Box plots of event rate and I_b/I_0 from Au nanospheres with 500 nm light and without light. Box plots show median (black line), mean (black square box), quartiles (boxes) and range (whiskers). Mann–Whitney U tests, *** $p < 0.001$.

For the analysis of the events, we have tuned the scale the same (updated Fig. S28 and S31) as suggested by the reviewer. Only for the data from 9h, a larger scale has to be used because the values are much larger.

Additionally, we would like to emphasize that the analysis based on scatter plots of analyte (t_d vs. I_b) are no longer working well in distinguishing Au nanoparticles, nanoplates and their aggregates in this system. Actually, we focused on the peak shape of the events rather than the translocation time and blocking current.

(1) When you use different sizes of nanopores for the reacted solutions stopped at different time, smaller aggregations will be ignored by larger pore sizes. How could you compare the degree of your aggregations?

Authors' response: We are very grateful for the comments. As we mentioned before, nanoparticles are always aggregating throughout the formation of nanoplates. Thus, the smallest aggregates should be at the beginning of the reaction, and the ~ 20 nm nanopores we used can completely cover the diameter of the aggregates at this stage. After 9 h of reaction, the size difference between nanoparticles and nanoplates in the aggregates is the largest. As we have demonstrated, the diameter of the smallest nanoparticle in the sample is about 35 nm, and we can detect these nanoparticle translocation events using a 120 nm nanopore. The nanopore size was chosen by the size distribution of nanostructures from TEM analysis to ensure that it can cover the range of the particle size.

Fig. R1 TEM images (a) and histogram (b) of the size distribution of nanostructures from 9 h sample. Pie chart of nanoparticles and nanoplates percentage content from TEM and nanopore (c). The TEM results contain 319 samples and the nanopore results contain 1198 events. I-t trace and corresponding TEM image and size distribution of 35 nm Au nanoparticles (d).

Fig. R2 TEM images of certain aggregations with less anisotropic morphologies

It is worth to mention that there may be deviation for the percentage analysis caused by certain aggregations with less anisotropic morphologies. For example, as shown in the

Fig. R2, these aggregates may not produce a distinct two-peak shape during nanopore detection, because the size difference of the two domains are too large or the coalescence is almost finished. In this case, the degree of detected aggregations may be lower than the real status. To reduce the deviation, we always collected more than 1000 events. Nevertheless, the deviation will not change the conclusion of the coalescence-induced growth of Au nanoplates.

(2) Since you have applied a 5 kHz filter (I guess a low-pass filter) to identify the events, then the events faster than 0.2 ms in a lot of your measurements (i.e. Figure S16, S21b, S22d to name a few) cannot be trustable at all. Although the event ratio seems correlated with TEM observation but once you have all the events below 0.2 ms removed, what is going to happen?

Authors' response: We are very grateful for the comments. All the electrical signals were detected with a 10 kHz four-pole Bessel low-pass filter (Methods), and then the events longer than 0.1 ms are reliable. In the case of Au seeds, the minimum dwell time of events in Figure S16 and S21b are 0.15 and 0.11 ms, respectively. The minimum dwell time in the 13 nm gold sphere translocation event is 0.18 ms (Figure S22d). The events are slower than 0.1 ms in our all measurements.

(3) Was Figure S31a without light illumination the same as Figure S16? Then what is the pie chart with 500nm illumination going to be?

Authors' response: We are very grateful for the comments. No, the data in these figures come from different nanopores. As shown in the pie chart below, the single-peak event rate increases slightly after the addition of light.

(4) For 5 nm gold spheres the average translocation time of 0.372 ms was longer than that (0.1 ms) of 13 nm gold spheres in Figure S21b. Is this reasonable?

Authors' response: We are very grateful for the comment. We apologize for the oversight in the dwell time of the 13 nm Au spheres and 5 nm Au spheres after centrifugation, which was demonstrated as the full width half maximum of the ionic current signature for each translocation event. We have now updated graphs in the revised manuscript. The average translocation times of these nanoparticles are shown in below. The average translocation time of 13 nm was longer than that of 5 nm Au spheres.

2. For Q7 R2, the authors did not correlate the references with my question. *Anal. Chem.*, 2016, 88, 9251-9258 and *Biophys. J.*, 2013, 105, 776-782 were simulation works. Following this, what is the predicted average translocation time considering the surface charge density of your gold spheres and the shape of your nanopores? *Nat. Nanotechnol.*, 2010, 5, 160-165 was the experimental work on (highly charged) DNA through SiN nanopores. How would this apply to your case?

Authors' response: We are very grateful for the comments. The complex nature of translocation dynamics with salt gradient makes it difficult to predict the translocation time of nanoparticles, which is different with the established model for predicting the translocation time of conical nanopore (Sexton, L., et al. *J. Am. Chem. Soc.* **2010**, 132, 6755–6763). The literatures (including SiN_x nanopores and conical nanopipettes) on the mean dwell time of Au particles translocation also varies widely, from less than one millisecond to tens of milliseconds. (Karhanek, M., et al. *Nano Lett.*, **2005**, 5, 403–407, German, S. R., et al. *J. Phys. Chem. C*, **2013**, 117, 703–711, Wu, H., et al. *J. Phys. Chem. C*, **2014**, 118, 26825–26835, Lin, X., et al. *Chem. Sci.*, **2017**, 8, 3905-3912, Karmi, A., et al. *ACS Appl. Nano Mater.*, **2020**, 4, 1000-1008.)

It has been reported experimentally (Wanunu, M., et al., *Nat. Nanotechnol.*, **2010**, 5, 160-165; Leong I. W., et al., *Anal. Chem.* **2021**, 93, 16700–16708; Charron, M. et al., *ChemRxiv* **2021**, 1–34.) and theoretically (Hatlo M. M, et al. *Phys. Rev. Lett.*, **2011**, 107, 068101; He. Y, et al. *Biophys. J.*, **2013**, 105, 776-782; Hsu, W. L., et al. *Anal. Chem.*, **2016**, 88, 9251–9258) that the salt gradient approach helps to manipulate the translocation dynamics through a reinforced electric field and induced electroosmotic flow, which not only slows down the motion of the object such as DNA (Wanunu, M., et al., *Nat.*

Nanotechnol., **2010**, 5, 160-165; Charron, M. et al., *ChemRxiv* **2021**, 1–34) and nanoparticles (Leong I. W., et al., *Anal. Chem.* **2021**, 93, 16700–16708) but also raise the capture rates. We assume that this simulation (Hsu, W. L., et al. *Anal. Chem.*, **2016**, 88, 9251–9258) is more suitable to explain our experimental system, and that the use of conical pores and salt concentration gradients can enhance the event rate and translocation time resolution.

REVIEWERS' COMMENTS

Reviewer #2 (Remarks to the Author):

My concerns are addressed. There are no additional major shortcomings.